# Systematic Review: Strategies for Improving HIV Testing and Detection Rates in European Hospitals

**DOI:** 10.3390/microorganisms12020254

**Published:** 2024-01-25

**Authors:** Klaske J. Vliegenthart-Jongbloed, Marta Vasylyev, Carlijn C. E. Jordans, Jose I. Bernardino, Silvia Nozza, Christina K. Psomas, Florian Voit, Tristan J. Barber, Agata Skrzat-Klapaczyńska, Oana Săndulescu, Casper Rokx

**Affiliations:** 1Section Infectious Diseases, Department of Internal Medicine, Erasmus University Medical Center, 3015 CN Rotterdam, The Netherlands; k.vliegenthart-jongbloed@erasmusmc.nl (K.J.V.-J.); m.vasylyev@erasmusmc.nl (M.V.); 2Astar Medical Center, 79041 Lviv, Ukraine; 3Department of Medical Microbiology and Infectious Diseases, Erasmus University Medical Center, 3015 CN Rotterdam, The Netherlands; c.jordans@erasmusmc.nl; 4HIV and Infectious Diseases Section, Department of Internal Medicine, Hospital Universitario La Paz-Carlos III, IdiPAZ, 28029 Madrid, Spain; jose.bernardino@salud.madrid.org; 5Centro de Investigación Biomédica en Red de Enfermedades Infecciosas, CIBERINFEC, 28029 Madrid, Spain; 6Department of Infectious Diseases, Università Vita Salute IRCCS San Raffaele Scientific Institute, 20132 Milan, Italy; nozza.silvia@hsr.it; 7European Hospital of Marseille, 13003 Marseille, France; kcpsomas@gmail.com; 8Department of Internal Medicine II, University Hospital Rechts der Isar, School of Medicine, Technical University of Munich, 80333 Munich, Germany; florian.voit@mri.tum.de; 9Department of HIV Medicine, Royal Free Hospital, London NW3 2QG, UK; t.barber@nhs.net; 10Institute for Global Health, University College London, London WC1E 6BT, UK; 11Department of Adults’ Infectious Diseases, Hospital for Infectious Diseases, Medical University of Warsaw, Wolska Street 37, 01-201 Warsaw, Poland; agata.skrzat@op.pl; 12Department of Infectious Diseases I, Carol Davila University of Medicine and Pharmacy Bucharest, No. 1 Dr. Calistrat Grozovici Street, 021105 Bucharest, Romania; oana.sandulescu@umfcd.ro; 13National Institute for Infectious Diseases “Prof. Dr. Matei Balș”, No. 1 Dr. Calistrat Grozovici Street, 021105 Bucharest, Romania

**Keywords:** undiagnosed HIV, AIDS, late diagnosis, Europe, HIV indicator conditions, HIV screening, HIV screening strategies, HIV test strategies

## Abstract

Undiagnosed HIV infection is a prominent clinical issue throughout Europe that requires the continuous attention of all healthcare professionals and policymakers to prevent missed testing opportunities and late diagnosis. This systematic review aimed to evaluate interventions to increase HIV testing rates and case detection in European hospitals. Out of 4598 articles identified, 29 studies fulfilled the selection criteria. Most of the studies were conducted in single Western European capital cities, and only one study was from Eastern Europe. The main interventions investigated were test-all and indicator-condition-based testing strategies. Overall, the prevalence of undiagnosed HIV was well above 0.1%. The studied interventions increased the HIV testing rate and the case detection rate. The highest prevalence of undiagnosed HIV was found with the indicator-condition-driven testing strategy, whereas the test-all strategy had the most profound impact on the proportion of late diagnoses. Nevertheless, the HIV testing rates and case-finding varied considerably across studies. In conclusion, effective strategies to promote HIV testing in European hospitals are available, but relevant knowledge gaps regarding generalizability and sustainability remain. These gaps require the promotion of adherence to HIV testing guidelines, as well as additional larger studies representing all European regions.

## 1. Introduction

Human immunodeficiency virus (HIV) warrants the continued attention of healthcare professionals and policymakers in Europe to prevent missed opportunities and facilitate timely diagnosis. Over the course of the last three decades, over 2.4 million people have been diagnosed with HIV in the European region as defined by the World Health Organization (WHO) [1]. With a prevalence of around 0.5%, HIV is still a serious continental public health concern, with its epicenter in Eastern Europe [2]. The United Nations’ aim of achieving a 75% reduction in new HIV infections in Europe was not reached. Between 2010 and 2020, Western/Central Europe showed an 11% decrease in new infections, while Eastern Europe showed a 43% increase [3]. The European Centre for Disease Prevention and Control (ECDC) estimated that 14% of persons living with HIV in Europe are still unaware of their diagnosis [2]. Additionally, 51% of those newly diagnosed with HIV presented late in the disease, with a CD4+ T-cell count of <350 cells/mm^3^ [1]. These numbers are concerning, given that late diagnosis is linked to increased rates of illness [4,5], mortality [6,7], costs [8,9], and HIV transmission [10].

HIV testing is offered in both healthcare and non-healthcare settings [11]. It is well described that HIV testing rates among those eligible vary by the testing venue used [12]. In hospitals specifically, the testing rates showed a wide distribution and differences among the emergency department (4–66%), inpatient department (17–73%), and outpatient department (35–98%) [12]. While aggregated European data are lacking, in the Netherlands, approximately one in three new HIV diagnoses occur within hospital settings [13]. The hospital setting provides an opportunity for testing individuals when symptoms and signs associated with HIV that are indicative of potential indicator conditions or risk factors for HIV are identified [14]. However, even in the presence of an indicator condition, HIV diagnosis could be missed because of insufficient awareness among healthcare professionals in hospitals about the need to offer an HIV test [15]. Ensuring the utilization of all testing opportunities is crucial to promote improved health by facilitating proper diagnosis that enables access to care and initiation of HIV treatment.

In the WHO’s European region, multiple HIV testing guidelines exist to facilitate appropriate HIV testing in the right individuals presenting to hospitals with indicator conditions or known risk factors (Table 1). The ECDC and WHO both recommend testing all persons with indicators for HIV and promote universal testing in settings and populations with a high HIV prevalence [16,17]. Universal opt-out testing strategies can also be recommended by national guidelines from high-income countries, e.g., as the British guidelines do when the HIV prevalence exceeds 2/1000 individuals in a region [18].

Despite clear guidelines, adherence to HIV testing guidelines in healthcare settings proved to be poor in the United Kingdom, with test coverage of only 27%, which was due to the low provider test offer as opposed to low patient acceptance [19]. Health service providers throughout Europe have highlighted gaps in HIV testing guidelines related to specific settings and population groups, as well as in the process of offering an HIV test [20]. A recent pan-European analysis revealed that still less than 50% of the guidelines on HIV indicator conditions included specific HIV testing recommendations [21]. These observations, taken together, can likely explain the inadequate HIV testing rates in hospitals.

A push forward for healthcare professionals in indicator-condition-based HIV testing was provided by the EuroTEST initiative in 2014 by publishing the practical guideline “HIV Indicator conditions: Guidance for Implementing HIV testing in Adults in Health Care Settings” [14]. This document strongly recommends HIV testing in any person presenting with an indicator condition associated with an undiagnosed HIV prevalence of >0.1%. The evidence base, including feasibility and effectiveness for the listed conditions, was provided by two large multicenter studies called the HIV Indicator Diseases across Europe Study (HIDES) 1 and HIDES 2 [22,23]. However, a systematic review conducted in Western countries revealed a low HIV testing rate of less than 50% in clinical settings with selected common indicator conditions [15]. These results indicate an important implementation gap [24].

Emphasizing the role of healthcare professionals in hospitals in facilitating timely HIV diagnoses is a priority, given their crucial role in the HIV care continuum to prevent late-stage disease and reduce transmission. Therefore, the objective of this systematic review was to identify the strategies employed in European hospitals to help healthcare professionals to enhance HIV testing rates and the detection of HIV cases, as well as to evaluate their efficacy.

## 2. Materials and Methods

### 2.1. Selection Criteria

A systematic search of the literature was conducted using Medline, Embase, and Web of Science from inception to 4 October 2023. Predefined inclusion and exclusion criteria were established based on the Population, Intervention, Comparison, Outcome (PICO) framework, outlined in Appendix A. Regarding population, the studies needed to be hospital-based; those conducted in primary care units, community services, or dedicated clinics for sexually transmitted infections (STIs), tuberculosis (TB), or antenatal care (ANC) were excluded. For the intervention, the study’s primary objective needed to evaluate intervention strategies aimed at improving HIV testing rates or HIV case-finding. Studies solely establishing prevalence rates, focusing primarily on other outcomes like acceptability, or examining the effects of passive system changes were excluded (for example, retrospective assessments of guideline changes without active implementation). All study types published after peer-review in journals were considered for inclusion, excluding reviews and conference abstracts.

### 2.2. Search

The detailed search, available in Appendix A, employed key terms including “HIV testing” or “HIV infections” coupled with terms indicating hospitals and the 53 individual countries in the WHO’s European region [25]. The screening process was conducted using Rayyan. It followed a two-step approach involving an initial assessment of titles and abstracts followed by a full-text evaluation of selected eligible articles. Discrepancies in article selection between the two inter-blinded reviewers (K.J.V.-J. and M.V.) were resolved through discussion with a third reviewer (C.R.) until consensus. The reference lists of the identified articles were cross-checked for relevant articles missed by the search.

### 2.3. Data Extraction and Presentation

Two reviewers (K.J.V.-J. and M.V.) individually extracted data from half of the selected studies using a standardized data extraction table. Data entry was cross-validated by the reviewers cross-checking one another’s findings. The data were categorized by the main interventions studied. A test-all strategy was considered to be the primary intervention if combined with other interventions because of the impact on the number of eligible persons. Data were collected on the setting, HIV testing strategy, and consent mode. Additionally, we delineated the testing continuum, comprising the entire population, the eligible population for an HIV test, HIV testing offer, test acceptance, and test rate (defined as the number of tests performed as a proportion of the eligible population). Both total and new diagnoses were tabulated, and the HIV detection rate was defined as the number of new HIV diagnoses as a proportion of the tested population. Comparisons consisted of changes in HIV testing rate or changes in HIV case-finding, in contrast to either a control group or baseline measurements. HIV case-finding was defined as the number of newly diagnosed HIV infections as a proportion of the entire studied population. The PRISMA guidelines were used for reporting [26] (Appendix A).

### 2.4. Quality Assessment

Quality assessment was performed using the Risk of Bias in Non-randomized Studies of Interventions (ROBINS-I) tool for non-randomized studies [27,28,29] and the Revised Cochrane Risk-of-Bias tool (RoB 2) for randomized studies [30,31]. Two independent reviewers (K.J.V.-J. and M.V.) evaluated each study, assessing seven domains. To ensure the accuracy and reliability of the quality assessment, a cross-checking process was conducted by two additional reviewers (C.C.E.J. and O.S.). Discrepancies identified during this phase were resolved through discussion with the first author.

## 3. Results

The initial search yielded a total of 4598 articles, resulting in 2702 records after duplicates were removed (Figure 1). Following the screening of titles and abstracts, 103 articles were selected for full-text review. After the application of the inclusion and exclusion criteria and the exclusion of a duplicate publication, 27 studies were included in the qualitative synthesis. Two studies were added by cross-reference checking, making a total of 29 studies [32,33].

### 3.1. HIV Testing Strategies

Table 2 summarizes the settings and HIV testing strategies described in the 29 included studies. The studies were published between 2012 and 2023. The majority of the studies were conducted in the United Kingdom (n = 14) [34,35,36,37,38,39,40,41,42,43,44,45,46,47], followed by France (n = 4) [48,49,50,51], Italy [52,53], Spain [32,54], the Netherlands [33,55], and Ireland [56,57] (each n = 2), while the other countries (Poland [58], Portugal [59], and Switzerland [60]) contributed one study each. The study settings primarily revolved around capitals or major teaching hospitals. The studies were conducted in the emergency department (ED; n = 14) [32,34,35,41,42,48,49,50,54,55,56,57,59,60], inpatient department (IPD; n = 8) [33,37,39,40,45,46,52,53], outpatient department (OPD; n = 5) [37,43,44,47,51], or a combination of OPD with ED or IPD (n = 2) [36,58]. The primary strategy studied was a test-all strategy (n = 15), of which over half were conducted in the ED (n = 9) [32,34,35,48,49,55,56,57,59], followed by IPD (n = 3) [38,39,40] or OPD settings (n = 3) [36,37,58]. Another key strategy used was indicator-condition-based testing (n = 8) [33,43,44,45,46,52,53,54]. The remaining studies primarily evaluated key-population-targeted testing (n = 2) [51,60], organizing a facility for rapid testing in the OPD (n = 1) [47], nurse-based point-of-care testing (n = 1) [50], or plan–do–check–act (PDCA) learning cycles (n = 2) [41,42]. Many studies consisted of multiple strategies. Table 3 summarizes the HIV testing rates and outcomes regarding case-finding for the different strategies.

### 3.2. Quality Assessment

Two randomized controlled trials showed minor and low concerns, respectively (Appendix A). The non-randomized studies had variations in the risk of bias (Appendix A). Overall, only 3 studies (11%) were categorized as having low risk, 15 (56%) showed moderate risk, and 9 (33%) showed serious risk of bias. Notably, the domains patient selection and deviations from intended interventions posed significant biases, indicating potential limitations in representativeness and adherence rates within the eligible population. Conversely, the measurement of outcomes and reporting of results were straightforward, with HIV tests as the primary method (Appendix A).

### 3.3. Test-All Strategy

ED test-all strategy studies had variable HIV testing rates, ranging from 3.9% in a study in France [48] to 89% in a study in Portugal [59]. In IPD settings, HIV testing rates ranged from 22% to 46% [38,39,40]. Six studies reported changes compared to a control group or baseline data [32,37,39,40,58,59]. Where evaluated, testing rates significantly increased [32,37,40], resulting in a rise from 0.5 to 2.6% in a Spanish ED [32], from 3 to 45% in an OPD in London for returning travelers [37], and from 4 to 23% among acute medical admissions in Leicester [40]. The ED test-all strategy found HIV detection rates ranging from 0.06% in the Netherlands and England to 0.86% in France [34,48,55]. IPD test-all strategies reported detection rates ranging from 0.42% to 2.2% [38,39,40,58]. Three studies were able to evaluate case-finding [32,40,59]. In EDs, a test-all strategy combined with either an electronic prompt or a nurse-driven test offer increased the case-finding in Portugal (from 13 to 27 per 100,000) [59] and Spain (from 3.2 to 22 per 100,000) [32]. Similar increases were observed at an IPD in England [40].

### 3.4. Indicator-Condition-Based Testing

Indicator-condition-based testing was assessed in eight studies. Where reported, the HIV testing rates increased [33,44,45,54]. Only one study was conducted in the ED, on 1.8 million attendees of 34 Spanish EDs, showing a 42% HIV testing rate after the implementation of an intensive training program [54]. Education and feedback were used to increase indicator-condition-based HIV testing in the Netherlands, reaching a testing rate of 37% [33]. Two studies used electronic prompts to promote indicator-condition-based testing, which resulted in increased HIV testing rates in people admitted to the ICU with pneumonia (from 28% to 73%) [45] and in people attending the OPD (from 3.2% to 34%) [44]. An intervention to enhance HIV testing among patients with neurological indicator conditions [46] or cervical dyskaryosis [43] resulted in testing rates of 8.7% and 46%, respectively, among the eligible population. The HIV detection rates were generally higher than with test-all strategies and were all well above 0.1% [33,52,53,54]. However, in four small studies, no new cases were identified [43,44,45,46]. Two Italian studies in IPDs found HIV detection rates of 3.7% and 3.8%, respectively [52,53]. HIV detection rates were lower in a Dutch IPD setting (0.2%) [33] and rose from 0.4% to 1.4% in the large Spanish ED study (1.4%) [54].

### 3.5. Other Strategies

Two studies focused on targeted testing of key populations and found a moderate test acceptance of around 50% [51,60], with a 1.8% HIV detection rate amongst migrants in a Parisian OPD [51]. Nurse-driven strategies by risk assessment and rapid testing [50], or by having nurse practitioners aid with indicator-based testing [32], increased the testing rates. Although almost all interventions incorporated teaching efforts, only three studies used teaching as a primary intervention, which led to increased HIV testing rates when used in indicator condition strategies in the abovementioned Spanish and Dutch studies [32,33], but not in a Polish study with a test-all strategy [58]. Plan–do–check–act (PDCA) cycle interventions in two ED-based studies increased testing rates and found 0.30% and 0.08% HIV prevalence, respectively, after implementing the addition of blood testing and incorporation of nursing staff into the HIV testing services [42], and through teaching, appointing an HIV advocate nurse, and adding HIV testing to a predefined care set [41]. Offering volunteer point-of-care testing in the OPD or ED also helped with the HIV case identification [47,50].

### 3.6. Late Diagnosis

In the ED, implementing universal HIV screening resulted in a reduction in the number of late presenters from 78% to 39% among the total diagnoses [59]. This was also observed among inpatients, where the percentage of late presenters among screened patients was 52%, as opposed to 92% among those who were tested within a targeted approach [39]. In Italian IPD settings using targeted testing, late presentation rates of around 75% were reported [52,53].

## 4. Discussion

In European hospital settings, the test-all and indicator-condition-based strategies were the most frequently studied interventions, and they successfully increased both HIV testing rates and HIV case detection. Nurse-delivered services, point-of-care testing, and PDCA cycles also showed positive effects. Electronic prompts, utilization of nurses, and educational activities were mostly used to support other strategies.

All controlled studies showed an increase in the HIV testing rates following the intervention [32,33,37,40,41,42,44,45,50,54], and nearly all of them reported increased HIV case-finding [32,40,50,54,59]. In all but one study, the increases were the result of a multifaceted intervention including PDCA, point-of-care testing, electronic prompts, and/or nurse assistance. However, the improvement attained in HIV testing rates was generally moderate, at around or below 50%. Poor testing rates (9–42%) were observed on many occasions, regardless of the setting, even when using well-defined indicator conditions [44,46,54]. Studies reported time pressure [42,50], lack of additional staff [48], requirement of explicit consent [48,54], lack of automated support [32,50], turnover of staff [46], HIV stigma [46], variability in uptake among consultants [46], and the clinician’s assessment that testing was not appropriate or not clinically indicated [44] as reasons for not testing. Despite the modest testing rates, healthcare professionals found satisfaction in the evident rise [54], the cost-effectiveness [50,54], or the success achieved in real-life scenarios [39,42,48,50]. Despite the availability of these clinically effective strategies, their suboptimal implementation perpetuates the existence of missed opportunities to test for HIV in European hospitals.

### 4.1. Test-All Strategy

The test-all strategy is arguably the strategy that could lead to the fewest missed opportunities. However, this strategy requires the availability of ample resources and personnel to execute it. Moreover, the HIV prevalence in the community should be high enough for a favorable cost–benefit ratio. Most of the test-all studies in EDs were conducted in capital cities in Western Europe, with reported community prevalences ranging from >1/1000 in Paris [48] and >2/1000 in Dublin and the United Kingdom [35,56,57] to 5.4/1000 in Wandsworth, London [34]. The HIV prevalence rates reported in the test-all studies in EDs were all above the 1/1000 (0.1%) cutoff point for cost-effectiveness [61,62,63]. However, the HIV detection rate was sometimes lower [34,55,56,57]. This signals a barrier for populations with undiagnosed HIV to present in hospital settings or to accept testing. The solution would be to implement more targeted approaches, or to reevaluate cost-effectiveness considering the added benefit of relinkage to care [34,56,59] and the necessity of accepting higher costs to identify new infections in low-prevalence settings.

### 4.2. Indicator-Condition-Based Testing

The highest prevalence of undiagnosed HIV was found with an indicator-condition-driven testing strategy [33,52,53,54]. This is a logical consequence of selecting people with a higher pretest likelihood of having HIV. Where assessed, the implemented interventions increased the HIV testing rates [33,44,45,54]. However, the selected indicator conditions varied; one study added chemsex and post-exposure prophylaxis [54], and one study included all indicator conditions [53]. While identification was based on diagnostic coding in Spain [54] and the Netherlands [33], manual case review was conducted in neurological inpatients [46] and patients attending the OPD [44] in England. The selection of the indicator conditions can have a profound impact on the reported HIV testing, with a recent meta-analysis showing variable testing rates with TB (72%), viral hepatitis (45%), malignant lymphoma (35%), and cervical carcinoma and dysplasia (12%) [15]. Moreover, the indicator conditions differ, with lower (pneumonia or seborrheic dermatitis) or higher (STI or viral hepatitis) HIV prevalence rates [14]. The selected indicator conditions and limited study size can prevent case identification [43,44,45,46]. Given the high yield and favorable cost–benefit ratio, indicator-condition-driven testing practices should be promoted in Europe as a minimum standard.

### 4.3. Consent Procedures

The methods for obtaining consent for HIV testing varied between studies. Some studies implemented an opt-out strategy [34,35,36,41,45,50,53,56,59], while others required written informed consent [48,49,53,55]. Additional procedures may be applicable due to the specific research context. Currently, EuroTEST is in the process of assessing standard practices concerning consent procedures [64]. It is noteworthy that written informed consent poses a significant barrier to HIV testing [65,66]. Encouragingly, a recent rapid guidance issued in the UK has introduced the concept of assumed consent for opt-out testing in emergency departments [67]. International guidelines emphasize that consent for HIV testing should align with that for any other medical test, removing the necessity for written consent [17,68].

### 4.4. Late Diagnosis

A mutual goal of all testing efforts is to prevent late HIV diagnosis [69]. Universal testing is most effective to decrease the proportion of late diagnosis. Although hospital admission for an HIV indicator condition is linked to late diagnoses, clinicians are still accountable for detecting HIV. Neglecting this responsibility prevents the patient from initiating antiretroviral therapy, leading to disease progression. Hence, testing efforts in IPDs should ensure that no person admitted with indicator conditions remains untested for HIV.

### 4.5. Research Gaps

The common clinical gap deduced from the data in this systematic review is in the evidence base regarding the effective implementation of intervention strategies and their generalizability to other currently unstudied settings. First, no interventional studies have been conducted across hospitals in multiple European countries. Second, a major underrepresented region in all European research initiatives in this field has been Eastern Europe, despite its significant burden of disease. Third, attention should be given to implementation science factors within the study design. Fourth, impact assessments would be improved by adequate control data and long-term follow-up, necessary to evaluate the sustainability of interventional programs. Fifth, data are needed on the prioritization of indicator conditions used in testing strategies when human and financial resources are limited. Sixth, a comprehensive assessment of cost-effectiveness is necessary, encompassing not just the direct cost of the test but also factoring in the unrewarded contributions of on-site nurses and the deployment of research assistants. Ideally, cost-effectiveness evaluations should result in a compelling business case that supports the intervention’s sustainability and gains acceptance from management. Intelligent machine learning tools for case identification have the potential to significantly impact this scenario. Lastly, nurse-based interventions showed promising results [32,50] and in supporting project-based strategies [41,42], ED-based test-all studies [48,49], and specialized clinics [37]. This signals relevant synergy between doctors and nurses that should be further studied when implementing HIV testing strategies.

### 4.6. Limitations

Several limitations should be acknowledged. Most of the studies had a truncated methodological quality, with only two being randomized in design and many lacking comparator groups. This limitation affects the overall strength of the evidence and introduces the risk of selection bias and deviations from the intended interventions. Furthermore, the heterogeneity in healthcare systems, populations, and HIV prevalence rates makes generalizing statements regarding the effectiveness of the strategies challenging. In relation to this point, the geographic diversity of the studies considered was limited.

## 5. Conclusions

In conclusion, this comprehensive analysis of HIV testing strategies in European hospitals revealed a nuanced landscape marked by diversity in approaches and outcomes. Effective strategies to promote HIV testing in hospitals exist but are inconsistently applied. There is a critical need for further research and inclusive implementation to optimally address undiagnosed HIV cases in Europe. These efforts aim to bring solutions to the table of HIV testing strategies in Europe, necessitating international initiatives to address a similar global unmet need.

## Figures and Tables

**Figure 1 microorganisms-12-00254-f001:**
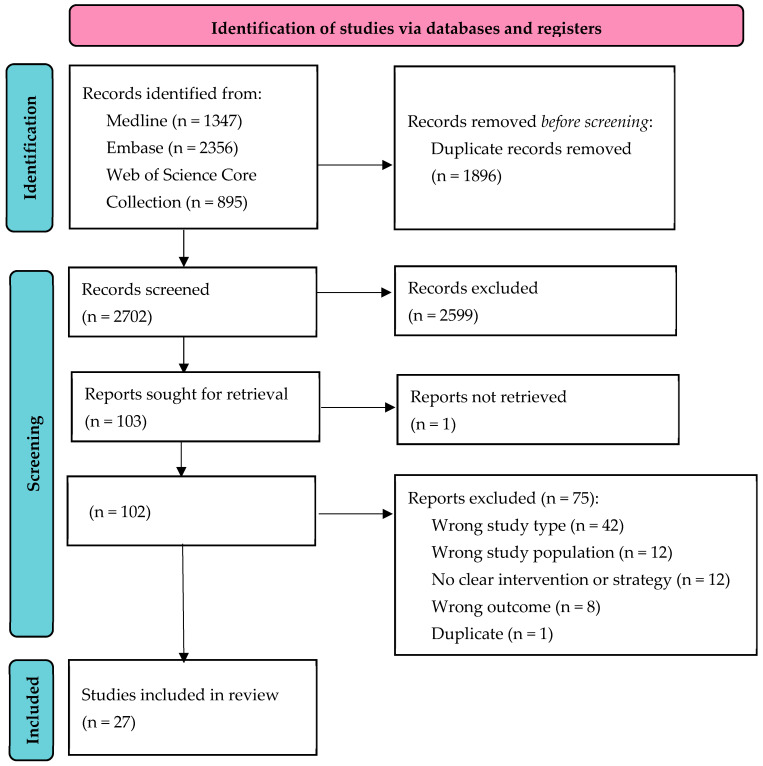
PRISMA 2020 flowchart for the inclusion of studies on HIV testing strategies in European hospitals.

**Table 1 microorganisms-12-00254-t001:** HIV test indications within the hospital setting based on European guidelines.

ECDC (2018) [16]	Offer integrated HIV/HBV/HCV testing to any person attending a hospital if they: ▪ Are known to be or identify as members of certain risk groups; ▪ Present with clinical symptoms suggestive of HIV infection; ▪ Present with an HIV indicator condition, including an STI, HBV, or HCV.
	Universal testing can be considered in geographical areas where the local diagnosed seroprevalence of an infection is high.
	The ECDC underlines that testing in hospital settings as well as routine testing in the emergency department is an acceptable strategy for patients and staff.
WHO (2021) [17]	In low-HIV-burden settings, offer HIV testing to: ▪ Adults, adolescents, or children who present in clinical settings with signs and symptoms or medical conditions that could indicate HIV infection, including TB, viral hepatitis, and STIs; ▪ HIV-exposed children and symptomatic infants and children; ▪ Key populations and their partners; ▪ All pregnant women. In high-HIV-burden settings, HIV testing should be offered to all populations and in all services. The WHO states that significant opportunities exist for integrating HIV testing into many clinical services, but that strategies should be guided by local epidemiology and HIV test coverage gaps.
BHIVA/ BASHH/ BIA (2020) [18]	Offer HIV testing to: ▪ People belonging to groups at increased risk of HIV exposure; ▪ Sexual partners of those diagnosed with HIV; ▪ People attending certain healthcare settings, i.e., for sexual health, addiction, antenatal care, or pregnancy termination; ▪ People commencing immunosuppressant therapy; ▪ People presenting with an HIV indicator condition in any healthcare setting; ▪ People presenting to any healthcare setting in areas of high HIV prevalence (2–5/1000) if undergoing venipuncture, or in areas of extremely high HIV prevalence (>5/1000) regardless of indication for venipuncture

Abbreviations: BHIVA/BASHH/BIA: British HIV Association/British Association for Sexual Health and HIV/British Infection Association; ECDC: European Centre for Disease Prevention and Control; WHO: World Health Organization; HIV: human immunodeficiency virus, HBV: hepatitis B virus, HCV: hepatitis C virus, STI: sexually transmitted infection, TB: tuberculosis.

**Table 2 microorganisms-12-00254-t002:** Summary of peer-reviewed studies on HIV testing strategies in European hospitals—descriptors and interventions.

Study			Setting			HIV Testing Strategy			
First Author	Year	Country, City	Setting + Sites	Population	Time Period	Intervention Category	Consent Mode *	Intervention Specification **	Study Design + Control Group
**Test-all:** Comprehensive testing approach aiming to screen all individuals presenting in a given setting, universal testing (ordered by: ED followed by OPD and IPD)									
Casalino [48]	2012	France, Paris	6 EDs	Adults (18–70 years)	12 months	Test-all POCT	Opt-in (WIC)	ANRS URDEP study: Routine HIV screening, using a rapid test on capillary blood. Training for testing and counselling as well as posters.	Cross-sectional Control: NA
d’Almeida [49]	2012	France, Paris	31 EDs	Adults (18–64 years)	16 months	Test-all POCT	Opt-in (WIC)	ED team training session (lecture, rapid test, practice of test, and counselling). Information sheet for patients. POCT by triage nurses and research assistant.	Cross-sectional Control: NA
Gómez- Ayerbe [32]	2019	Spain, Madrid	1 ED	Adults (18–60 years)	12 months	Test all Nurse	NS	DRIVE program. Inclusion in medical or nursing consultations. Trained nurse practitioners. Questionnaire on HIV IC and risk assessment. POCT.	Cross-sectional Historic control
Grant [56]	2020	Ireland, Dublin	1 ED	Adults (>17 years) already receiving a blood test	36 months	Test-all	Opt-out	Opt-out testing for HIV, HBV, and HCV	Cross-sectional Control: NA
Luiken [55]	2017	Netherlands, two cities	3 EDs	Adults (>17 years) already receiving a blood test	14 months	Test-all	Active (WIC)	Patients were informed by posters and flyers. HIV test with an extra blood sample. Anonymized batch testing of those not consenting.	Cross-sectional Control: NA
Marchant [34]	2022	England, London	1 ED	Adults (18–59; later 16+ years)	3 years	Test-all Prompt	Opt-in/opt-out	HIV testing was added to all ED blood test order sets.	Cross-sectional Control: NA
O’Connell [57]	2016	Ireland, Dublin	1 ED	Adults (>18 years) already receiving a blood test	10 months	Test-all	Opt-out	Opt-out testing for HIV, HBV, and HCV on an extra blood sample. Patients were informed by posters and leaflets in seven languages. Staff teaching.	Cross-sectional Control: NA
Orkin [35]	2016	England, Scotland	9 EDs	Adults (>17) already receiving a blood test	6 days	Test-all	Opt-out	“Going Viral” campaign. Opt-out testing for HIV, HBV, and HCV. Staff were informed by training and patients were informed by posters and leaflets.	Cross-sectional Control: NA
Vaz-Pinto [59]	2022	Portugal, Cascais	1 ED	Adults (18–64) already receiving a blood test	3 years	Test-all Prompt	Opt-out	Automatically generated HIV test request if exclusion criteria were not met (age, no bloodwork, known HIV-positive, or tested). Nurses’ training.	Cross-sectional Historic control
Bath [36]	2016	England, London	2 EDs/6 OPDs	Adults (>16 years) already receiving a blood test	5 days	Test-all Project	Opt-out	TestMeEast: HIV testing during National HIV Testing Week. Student and charity volunteers, training session, social media, banners, posters, leaflets.	Cross-sectional Control: NA
Herbert [37]	2012	England, London	1 OPD	Adults (>17) attending a returning traveler clinic	28 months	Test-all POCT	Active choice	Targeted vs. universal. Phase 0: symptom-based testing. Phase 1: universal offer of HIV test. Phase 2: POCT (15 months) + training.	Cross-sectional Baseline control
Cholewińska [58]	2020	Poland	4 OPDs/IPDs	Patients eligible for HIV test according to MD	6 months	Test-all Edu	NS	Nationwide project “STOP Late Presenters”: (a) voluntary training in the form of a presentation; (b) information materials and leaflets.	Cross-sectional Historic control
Burns [38]	2012	England, London	1 IPD	Adults (19–65 years) presenting to AMU.	16 weeks	Test-all POCT	Active choice	RAPID: Employment of a health advisor (HA) offering POCT with the aid of an educational video available in up to four languages.	Cross-sectional Control: NA
Hill-Tout [39]	2016	England, London	1 IPD	AMU targeted testing without vs. with screening	19 months	Test-all	NS	Targeted versus universal. Routine HIV screening was introduced in the acute medical unit (AMU). This study audited the effects retrospectively.	Cross-sectional Historic control
Palfreeman [40]	2013	England, Leicester	1 IPD	New admissions (15–59 years) admitted to the AMU	24 months	Test-all	Opt-in	Routine testing in the AMU, introduced to staff by e-mail/meetings and to patients by posters/flyers. Pilot phase: weekly AMU visit. Post-pilot: no visits.	Cross-sectional Pre-intervention
**Project:** Implementation of comprehensive projects, campaigns, or plan–do–check–act (PDCA) cycles aimed at promoting HIV testing.									
Fox [41]	2022	England, London	1 ED	Adults (16–59 years)	29 months	Project	Opt-in/opt-out	PDCA cycle: (1) survey for barriers; (2) teaching session for ED; (3) HIV advocate nurse champion; (4) Prompts; (5) Gamified teaching; (6) HIV testing to care set.	Cross-sectional Pre-intervention
Rayment [42]	2013	England, London	1 ED	Adults (16–65 years) Later: no age limit	30 months	Project	Opt-in	Implementation based on HINTS study: PDCA cycle, training, nurse-based testing, champions, incentivization, information technology solutions.	Prospective Cross-sectional
**IC (indicator-condition-guided testing):** Targeted testing of individuals based on medical conditions or symptoms that indicate a potential risk for HIV infection (ED; OPD; IPD).									
Gonzalez Del C. [54]	2023	Spain	34 EDs	People presenting with one of the six prioritized HIV ICs	6 months	IC Edu	NS	Intensive training program “dejatuhuella”, focused on testing in six HIV ICs. ^a^ Four educational sessions in every ED, along with webinars, courses, and meetings.	Cross-sectional Pre-intervention
Qureshi [43]	2017	England, Birmingham	1 OPD	Women with cervical dyskaryosis for colposcopy	21 months	IC	Active choice	Offer of HIV testing as part of clinical management. An information leaflet upon arrival at the clinic. Discussion of questions about HIV testing.	Cross-sectional Control: NA
Youssef [44]	2018	England, Brighton	3 OPDs	Patients aged > 15 years attending three specific OPDs	12 weeks	IC Prompt	NS	Singular education program followed by either prompt (6 weeks) or no-prompt (6 weeks). Prompt identified HIV ICs before appointment.	Non-randomized Crossover trial
Barbanotti [52]	2023	Italy, Milan	1 IPD	Admitted patients identified with an HIV IC in seven wards	24 months	IC	Opt-in	ICEBERG study: A dedicated healthcare professional in charge of patients’ enrolment and HIV test prescription in case of an observed HIV IC ^b^.	Cross-sectional Control: NA
Bogers [29]	2022	Netherlands Amsterdam	5 IPDs	Adults (>18) with HIV ICs in disease billing code	12 months	IC Edu	NS	PROTEST 2.0: HIV ICs were assessed using electronic health records. ^c^ Interventions: presentation, discussion, feedback, pocket cards, posters.	Cross-sectional Pre-intervention
De Vito [53]	2023	Italy, Sassari	1 IPD	Patients identified with an HIV IC in one of six wards	16 months	IC	Opt-in (WIC)	SHOT Project: Each ward was provided with forms to collect data from patients, included in the screening in case of an observed HIV IC ^d^.	Cross-sectional Control: NA
Sharvill [45]	2017	England, Bath	1 IPD	Adults admitted to the ICU with pneumonia	1 year	IC Prompt	Opt-out	Routine HIV testing was added to the automated pneumonia screen.Prompt in case of diagnoses, pneumonia, RTI, chest infection, or chest sepsis.	Cross-sectional Pre-intervention
Sokhi [46]	2015	England, Sheffield	1 IPD	Patients admitted as acute non-stroke neurology cases	2 years	IC Prompt	Opt-in	Four phases: (1) Protocol disseminated to clinical staff; (2) Protocol and posters on noticeboards/offices/trolleys; (3) Prompt and education; (4) Continuation phase.	Cross-sectional Control: NA
**Other HIV testing strategies** (ED; OPD).									
Gillet [60]	2018	Switzerland, Lausanne	1 ED	Adults (18–75 years)	3 months	Key vs. test-all	Active choice	Targeted arm: Testing offer based on HIV testing criteria. Non-targeted arm: Active choice based on information on HIV. Crossover to the other arm.	Randomized controlled study
Leblanc [50]	2018	France, Paris	8 EDs	Adults (18–64)	12 months	Nurse POCT	Opt-out	DICI-VIH study: ED randomization to symptom-driven physician testing alone or additional nurse-based POCT based on risk assessment. Crossover.	Cluster randomized trial
Aparicio [51]	2012	France, Paris	1 OPD	Adults from SS Africa, the Antilles, Réunion, and Guyana	28 days	Key	Opt-in	Targeted testing: Patients presenting with a medical problem, wound dressing, or blood sample were offered testing by the attending doctor/nurse.	Cross-sectional Control: NA
Freer [47]	2015	England, London	1 OPD	People deciding to visit a rapid HIV testing service	12 months	POCT	Active choice	Universal testing: A rapid walk-in HIV testing service with oral swabs in the OPD for people attending the OPD or walk-in. Information by posters and staff.	Cross-sectional Control: NA

***Definitions:* Universal:** An approach to offer HIV testing to an entire population, avoiding selection based on individual risk factors or features. **Targeted:** Focusing HIV testing efforts on specific groups identified based on features, risk assessment, or indicator conditions that suggest a higher likelihood of HIV infection. * Consent mode: **Opt-in:** Individuals are not automatically offered an HIV test. They must proactively request the test or give explicit consent for it. **Opt-out:** Automatic offering of an HIV test to eligible individuals, as a routine part of healthcare. Testing is performed unless the individual actively declines. **Active (active choice):** Individuals are presented with the advantages and disadvantages of an HIV test, which allows them to make an informed decision. **WIC (written informed consent):** Documented agreement in which an individual voluntarily and explicitly consents to receive HIV testing. ** Intervention specification (only the two main interventions of each paper are listed, based on the reviewer’s opinion): **POCT (point-of-care testing):** Utilization of immediate diagnostic tests conducted at the point of patient care to rapidly determine HIV status. **Edu (educational program or staff training):** Implementation of structured educational programs or training sessions for healthcare staff. **Prompt (electronic reminders):** Integration of electronic prompts or reminders within hospital systems to prompt healthcare providers for HIV testing opportunities. **Nurse (nurse-led or specific personnel testing):** HIV testing conducted by nurses, health advisors, or other designated healthcare personnel within the hospital. **Key (key group targeting):** Tailored testing strategies focusing on high-risk groups through risk assessment or comprehensive sexual history evaluations. **HIV ICs:** ^a^ Sexually transmitted infections (STI), community-acquired pneumonia (CAP) aged 18–65 years, mononucleosis-like syndrome (MNS), herpes zoster (HZ) aged 18–65 years, chemsex (CS), and HIV post-exposure prophylaxis (PEP); ^b^ herpes zoster <60 years, persistent herpes simplex, pneumonia < 50 years, mononucleosis-like syndrome, invasive pneumococcal disease < 50 years, candidemia, dementia < 60 years, cerebral lesions, persistent leukopenia, persistent thrombocytopenia, gonorrhea, hepatitis (A, B, or C), syphilis, anal carcinoma, Hodgkin’s lymphoma; ^c^ tuberculosis (TB), cervical cancer/high-grade cervical dysplasia, vulvar cancer/high-grade vulvar dysplasia, malignant lymphoma, hepatitis B (HBV), hepatitis C (HCV), and peripheral neuropathy; ^d^ defined by the ECDC; ***Abbreviations:* AMU** = acute medical unit; **ED = emergency department**; **HA** = health advisor; **HIV** = human immunodeficiency virus; **HIV IC** = HIV indicator condition; **HBV** = hepatitis B virus; **HCV** = hepatitis C virus; **ICU** = intensive care unit; **IPD = inpatient department**; **MD = medical doctor**; **NS =** not specified; **OPD = outpatient department**; **PDCA** = plan–do–check–act; **POCT** = point-of-care test; **RTI** = respiratory tract infection; **SS** = sub-Saharan.

**Table 3 microorganisms-12-00254-t003:** Summary of peer-reviewed studies on HIV testing strategies in European hospitals—results.

Study, Setting, and Strategy			HIV Testing Strategy				Control Group				Comparison
First Author, Year	Country, Setting	Strategy	Population		Testing Rate *	New HIV Diagnoses (HIV Detection Rate) **	Population		Testing Rate *	New HIV Diagnoses (HIV Detection Rate) **	(HIV Testing Rate and HIV Case-Finding)
			Eligible (Total)	Tested			Eligible (Total)	Tested			
**Test-all:** Comprehensive testing approach aiming to screen all individuals presenting in a given setting (ordered as follows: ED, followed by OPD and IPD).											
Casalino, 2012 [48]	France, ED	Test-all POCT	183,957 (311,153)	7215	3.9%	40 (0.55%)					
d’Almeida, 2012 [49]	France, ED	Test-all POCT	78,411 (138,691)	12,754	16%	18 (0.14%)					
Gómez-Ayerbe, 2019 [32]	Spain, ED	Test-all Nurse	NA (63,054)	1635	2.6%	14 (0.86%)	NA (63,054)	966	0.5%	1 (0.62%)	**Testing rate: increased:** 0.5% vs. 2.6%. **Case-finding: increased:** 3.2 vs. 22.2 per 100,000 ED visitors.
Grant, 2020 [56]	Ireland, ED	Test-all	88,854 (140,500)	41,535	47%	38 (0.09%)					
Luiken, 2017 [55]	Netherlands ED	Test-all	7577 (NA)	3223	43%	2 (0.06%)					
Marchant 2022 [34]	England, ED	Test-all	110,683 (NA)	78,333	70%	50 (0.06%)					
O’Connell, 2016 [57]	Ireland, ED	Test-all	18,819 (40,000)	8839	47%	7 (0.08%)					
Orkin, 2016 [35]	UK, ED	Test-all	7807 (NA)	2118	27%	6 (0.52%)					
Vaz-Pinto, 2022 [59]	Portugal, ED	Test-all Prompt	43,153 (252,153)	38,357	89%	69 (0.18%)	NA (282,751)	NA	NA	37	**Testing rate:** NA. **Case-finding: increased:** 13 vs. 27 per 100,000 ED visitors.
Bath, 2016 [36]	England, ED/OPD	Test-all Project	4317 (10,386)	2402	56%	3 (0.12%)					
Herbert ^1^, 2012 [37]	England, OPD	Test-all POCT	NA (3623)	1444	40%	9 (0.62%)	NA (1342)	38	2.8%	0	**Testing rate: increased:** Phase 0 vs. 1 vs. 2; 2.8% vs. 23% vs. 45%. **Case finding:** NA.
Cholewińska 2020 [58]	Poland, OPD/IPD	Test-all Edu	NA (NA)	869	NA	4 (0.87%)	NA (112,928)	878	0.8%	NA	**Testing rate:** NA. Denominator unknown. **Case-finding:** NA. Denominator unknown.
Burns, 2012 [38]	England, IPD	Test-all POCT	282 (606)	131	46%	3 (2.22%)					
**Test-all:** Comprehensive testing approach aiming to screen all individuals presenting in a given setting (ordered as follows: ED, followed by OPD and IPD)—continued.											
Hill-Tout ^2^, 2016 [39]	England, IPD	Test-all	NA (19,110)	4955	26%	21 (0.42%)	NA (NA)	NA	NA	88	**Testing rate:** NA. **Case-finding:** NA.
Palfreeman 2013 [40]	England, IPD	Test-all	5517 (NA)	938	17%	10 (1.07%)	5484 (NA)	205	3.7%	4 (1.95%)	**Testing rate: increased.** Pre-pilot vs. pilot vs. post-pilot: 3.7% vs. 17% vs. 22.5%. **Case-finding: increased.** 7 vs. 18 vs. 24 per 10,000 admissions.
			Post-pilot 6225	1399	23%	15 (1.07%)					
**Project:** Implementation of comprehensive projects, campaigns, or plan–do–check–act (PDCA) cycles aimed at promoting HIV testing.											
Fox, 2022 [41]	England, ED	Project	NA (46,375)	9600	21%	8 (0.08%)	NA (42,809)	2825	6.6%	NA	**Testing rate: increased.** From baseline to end: 8% to 44%. **Case-finding:** NA.
Rayment, 2013 [42]	England, ED	Project	44,582 (NA)	4327	9.7%	13 (0.30%)					**Increased testing rate:** months 1–22 to 22–30, 11% vs. 29%. **Case-finding:** NA.
**Other HIV testing strategies** (ED; OPD).											
Gillet ^3^, 2018 [60]	Switzerland ED	Key vs. test-all	17 (80)	8	10%	0	80 (80)	38	48%	0	**Testing rate: no increase.** Targeted versus universal approach: 10% vs. 48%. **Case-finding:** NA.
Leblanc, 2018 [50]	France, ED	Nurse POCT	74,161 (102,240)	2915	3.9%	22 (0.54%)	74,166 (105,582)	92	0.12%	6 (6.5%)	**Testing rate: increased.** Physician- vs. nurse-driven: 0.12% vs. 3.9%. **Case-finding: increased.** 0.8 vs. 3.0 per 10,000 ED visitors.
Aparicio, 2012 [51]	France, OPD	Key	272 (NA)	166	61%	3 (1.8%)					
Freer, 2015 [47]	England, OPD	POCT	NA (NA)	148	NA	3 (1.4%)	NA (NA)	420	NA	0	**Testing rate:** NA. **Case-finding:** NA.
**IC (indicator-condition-guided testing):** Targeted testing of individuals based on medical conditions or symptoms that indicate a potential risk for HIV infection (ED; OPD; IPD).											
Gonzalez Del Castillo, 2023 [54]	Spain, ED	IC Edu	16,618 (1,796,741)	7002	42%	224 (1.67%)	15,879 (1,670,027)	3393	21%	65 (0.93%)	**Testing rate: increased** among ED visitors (0.42% vs. 0.75%) and among HIV ICs (21% vs. 42%). **Case-finding: increased** among ED visitors (3.9 vs. 12 per 100,000) and among HIV ICs (0.41% vs. 1.35%).
Qureshi, 2017 [43]	England, OPD	IC	533 (3262)	244	46%	0					
Youssef, 2018 [44]	England, OPD	IC Prompt	215 (NA)	74	34%	0	252 (NA)	8	3.2%	0	**Testing rate: increased** among HIV ICs without prompt vs. with prompt (3.2% vs. 34%). **Case-finding:** NA.
Barbanotti, 2023 [52]	Italy, IPD	IC	NA (NA)	520	NA	20 (3.8%)					
Bogers, 2022 [29]	Netherlands IPD	IC Edu	1256 (NA)	590	47%	1 (0.2%)	6739 (NA)	2478	37%	17 (0.7%)	**Testing rate: increased** among HIV ICs (37% vs. 47%). **Case-finding: reduced** among HIV ICs.
De Vito, 2023 [53]	Italy, IPD	IC	NA (NA)	300	NA	11 (3.7%)					
Sharvill, 2017 [45]	England, IPD	IC Prompt	59 (NA)	48	81%	0	68 (NA)	22	32%	0	**Testing rate: increased,** 32% vs. 81%. **Case-finding:** NA.
Sokhi, 2015 [46]	England, IPD	IC Prompt	4349 (6723)	378	8.7%	0					

***Definitions and categories:*** Setting: **IPD (inpatient department)**, **OPD (outpatient department)**, **ED (emergency department)**. Intervention (only the two main interventions of each paper are listed, based on the reviewers’ opinion): **Test-all:** Comprehensive testing approach aiming to screen all individuals presenting in each setting. **POCT (point-of-care testing):** Utilization of immediate diagnostic tests conducted at the point of patient care to swiftly determine HIV status. **IC (indicator-condition-guided testing):** Testing individuals based on specific medical conditions or symptoms that indicate a potential risk for HIV infection. **Edu (educational program or staff training):** Implementation of structured educational programs or training sessions for healthcare staff to enhance their knowledge and capacity regarding HIV testing protocols. **Prompt (electronic reminders):** Integration of electronic prompts or reminders within hospital systems to prompt healthcare providers for HIV testing opportunities. **Project (hospital-wide campaigns):** Implementation of comprehensive projects, campaigns, or plan–do–check–act (PDCA) cycles aimed at promoting HIV testing across the hospital environment. **Nurse (nurse-led or specific personnel testing):** HIV testing conducted by nurses, health advisors, or other designated healthcare personnel within the hospital setting. **Key (key group targeting):** Tailored testing strategies focusing on high-risk groups through risk assessment or comprehensive sexual history evaluations. * HIV testing rate is defined as the number of HIV tests performed as a proportion of the eligible population (if not available, the total population). ** HIV detection rate is defined as the total number of new HIV diagnoses as a proportion of the tested population. ***Abbreviations:*** NA (not available): Information is not available, not applicable, or not accessible. vs. (versus). UK (United Kingdom). If numbers are expressed as “**n + n**”, this means that additional cases are tested or identified among the non-eligible population. ***Footnotes:***
^1^ Herbert et al.: Targeted testing (Phase 0) followed by universal offer of HIV testing (Phase 1) and introduction of POCT (Phase 2); ^2^ Hill-Tout et al.: Retrospective comparison between symptom-based targeted testing (Audit A) and a combination of screening and targeted testing (Audit B). In the intervention group (Audit B), diagnoses were made by screening as well as targeting; Gillet et al.: in the targeted approach, people completed a risk factor assessment and patients with risk factors were offered free rapid testing. In the universal (non-targeted) approach, people received information about HIV and testing and were then offered testing.

## Data Availability

The findings presented in this systematic review are based on a comprehensive analysis of previously published data, referenced in the accompanying bibliography. No new data were generated or included.

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
