# Peer review of "Systematic Review: Strategies for Improving HIV Testing and Detection Rates in European Hospitals"

_microorganisms, 2024, doi:10.3390/microorganisms12020254_

Round 1

Reviewer 1 Report

Comments and Suggestions for Authors

Firstly, congratulations on writing the manuscript. As in Europe, but throughout the world, widespread testing is extremely necessary to identify cases. The WHO target of 95% of individuals knowing their serological status is still far from being achieved in some places.

Overall, I think the manuscript is well prepared, with appropriate references, well-outlined methodology and discussion focused on the most important points.

In the results, I found the format in which tables 2 and 3 were presented confusing. The information contained in them was not so objective and easy to interpret. Perhaps summary tables that presented less information could be easier to understand between the differences presented in the references used in the review.

Author Response

Thank you for your review of our manuscript titled ‘Strategies for improving HIV testing and detection rates in European hospitals’.  We appreciate the time and effort you invested in providing detailed feedback. Your suggestions have been instrumental in enhancing the clarity of the article.
The preparation of table 2 and 3 posed challenges for our group, as our initial goal of completeness led to difficult to read tables. In alignment with your suggestion, we have prioritized essential content. We trust that these adjustments have enhanced the overall value of the manuscript.

Reviewer 2 Report

Comments and Suggestions for Authors

Dear Authors,

I read your Manuscript with great interest; I think this topic need more attention, and in this scenario, your systematic review is of great importance.

I suggest revising the tables for minor typos (ex. Netherlands, Amsterdam in Table 2; also, some lines are missing both in Table 2 and 3, I am not sure if it is a problem related to the change of format from word to pdf).

Regarding the content, the study from De Vito et al. is reported to have used an opt-out strategy, while we used an opt-in instead. Please, revise if all the other studies are reported correctly.

Comments on the Quality of English Language

The Manuscript is well written and easily readable. Only minor typos must be addressed.

Author Response

Thank you for reviewing our manuscript, 'Strategies for Improving HIV Testing and Detection Rates in European Hospitals.' We value the time and effort you dedicated to providing detailed feedback. The identified typos have been corrected. While the cause of missing lines in Tables 2 and 3 is unclear to us, we've heeded the suggestion from one reviewer and streamlined the content in both tables to focus on the most essential information. We hope this enhanced the clarity of the tables, especially table 3. We revised the study by De Vito et al and checked also all other included studies that they are reported correctly.

Reviewer 3 Report

Comments and Suggestions for Authors

Thank you for this interesting work showing the importance of diversity in testing approaches. The article gives a good overview, it is well structured and easy to read.   

Author Response

Thank you for your review of our manuscript titled ‘Strategies for improving HIV testing and detection rates in European hospitals’.  We appreciate the time and effort you invested.